# Highly Sensitive, Robust, and Recyclable TiO_2_/AgNP Substrate for SERS Detection

**DOI:** 10.3390/molecules27196755

**Published:** 2022-10-10

**Authors:** Hsing-Yu Wu, Hung-Chun Lin, Yung-Hsien Liu, Kai-Lin Chen, Yu-Hsun Wang, Yung-Shin Sun, Jin-Cherng Hsu

**Affiliations:** 1System Manufacturing Center, National Chung-Shan Institute of Science and Technology, New Taipei City 237209, Taiwan; 2Center for Astronomical Physics and Engineering, Department of Optics and Photonics, National Central University, Taoyuan City 320317, Taiwan; 3Department of Electro-Optical Engineering, National Taipei University of Technology, Taipei 10608, Taiwan; 4Department of Physics, Fu Jen Catholic University, New Taipei City 242062, Taiwan; 5Department of Chemical and Materials Engineering, Chung Cheng Institute of Technology, National Defense University, Taoyuan City 335009, Taiwan; 6Graduate Institute of Applied Science and Engineering, Fu Jen Catholic University, New Taipei City 242062, Taiwan

**Keywords:** surface-enhanced Raman spectroscopy, silver nanoparticles, direct-current sputtering, titanium dioxide, photocatalytic degradation, recyclability, limit of detection, enhancement factor, rhodamine 6G, paraquat, acetylcholine

## Abstract

Label-free biosensors provide an important platform for detecting chemical and biological substances without needing extra labeling agents. Unlike surface-based techniques such as surface plasmon resonance (SPR), interference, and ellipsometry, surface-enhanced Raman spectroscopy (SERS) possesses the advantage of monitoring analytes both on surfaces and in solutions. Increasing the SERS enhancement is crucial to preparing high-quality substrates without quickly losing their stability, sensitivity, and repeatability. However, fabrication methods based on wet chemistry, nanoimprint lithography, spark discharge, and laser ablation have drawbacks of waste of time, complicated processes, or nonreproducibility in surface topography. This study reports the preparation of recyclable TiO_2_/Ag nanoparticle (AgNP) substrates by using simple arc ion plating and direct-current (dc) magnetron sputtering technologies. The deposited anatase-phased TiO_2_ ensured the photocatalytic degradation of analytes. By measuring the Raman spectra of rhodamine 6G (R6G) in titrated concentrations, a limit of detection (LOD) of 10^−8^ M and a SERS enhancement factor (EF) of 1.01 × 10^9^ were attained. Self-cleaning was performed via UV irradiation, and recyclability was achieved after at least five cycles of detection and degradation. The proposed TiO_2_/AgNP substrates have the potential to serve as eco-friendly SERS enhancers for label-free detection of various chemical and biological substances.

## 1. Introduction

Biosensors are analytical devices used in the detection of chemical and biological substances, and fluorescence-based methods are routinely employed because of their high sensitivity, safety, and large selection of labeling agents such as fluorophores and quantum dots [1,2]. However, there are several drawbacks with regard to these “labeled” methods, including high cost, complicated and laborious labeling steps, photo-bleaching of fluorophores, difficulties in making comparisons among different fluorophores, and potential changes to binding kinetics in detecting biomolecular interactions [3,4]. To overcome these shortcomings, various optical techniques have been developed to serve as label-free biosensors. Examples include those based on surface plasmon resonance (SPR), interference, ellipsometry, and spectra. Although SPR and ellipsometry biosensors require no labels, they are limited only to surface-based detections. Moreover, the setup and components in these label-free platforms are usually complicated. By comparison, spectrum-based biosensors provide a simple and feasible platform for analyzing chemical and biological substances on surfaces and in solutions. First developed in 1928, Raman spectroscopy has been used in various fields, such as environmental monitoring, forensic science, material science, mineralogy, medical diagnostics, and surface analysis [5]. When the incident light (e.g., a laser light source) is scattered from a molecule, it shifts a little in wavelength. The spectrum depends on the structure of the molecule, with each peak corresponding to the vibration of a specific molecular bond (e.g., C–C, C=C, N–O, C–H…) or a group of bonds (e.g., benzene ring, lattice mode…). Later in 1974, Fleischmann et al. reported the first surface-enhanced Raman spectroscopy (SERS), where silver electrodes with high surface areas were prepared to detect two types of pyridine [6]. Such enhancement can be attributed to the combination of both electromagnetic (EM) and chemical (CE) mechanisms [7,8]. With a total signal enhancement of 10^10^–10^14^ for a single molecule, it was estimated that the contribution of the EM part, through the coupling between the molecule and the EM field, could be up to 10^8^ [9,10]. Therefore, the EM effect is believed to be an essential factor in increasing the sensitivity of SERS [11].

In SERS, the so-called hotspots within the metallic nanostructures are highly localized regions of the intense local EM field. These locations can be formed with nanostructured clefts, gaps, and fissures, and an interparticle distance of less than 10 nm is preferable for the enhancement [12,13]. Shiohara et al. reviewed the techniques designed for creating SERS hotspots both in solution and on solid supports, including spherical and anisotropic nanoparticles in solution, assembly of nanoparticles onto solid supports, and top-down approaches based on lithography techniques [14]. As mentioned, SERS provides a label-free and nondestructive platform for monitoring chemical and biological samples down to single molecules on surfaces and in solutions [14,15,16]. With the advantages of sensitivity, rapidity, and simplicity, SERS has been commonly and widely used in the fields of medical diagnosis [17], water containments [18], food safety [19], and biochemical analysis [20].

Therefore, preparing high-quality substrates with maximum enhancement is crucial to SERS applications. A big challenge is that these substrates quickly lose their stability, sensitivity, and repeatability when stored in the atmosphere, mainly due to oxidation and carbon contamination of silver-based substrates [21]. The SERS intensity decreased exponentially after the substrate was exposed to the ambient air, which consequently led to an increased standard deviation [21]. To be improved, the stability, wet chemistry-based methods are used to construct nanoparticles on SERS substrates [22,23]. However, these synthesis processes are usually time-consuming, and the surface morphology of resulting substrates is not reproducible. Techniques such as nanoimprint lithography [24], spark discharge [25], and laser ablation [26] are also applied to the fabrication of substrates, but these methods are either complicated or not reproducible in surface topography. Direct-current (DC) sputtering, with which conductive materials can be deposited onto the substrate, provides an effective and reliable way of fabricating SERS substrates. By accelerating ionized gas molecules to bombard the target, nanoscaled target molecules can be vaporized onto the substrate. For example, by combining co-sputtering with atomic layer deposition technology, Yin et al. developed a stable SERS substrate for glycerin detection [27]. The effects of annealing on the microstructure and SERS performance of DC magnetron-sputtered Mo–Ag substrates were investigated, indicating that the Ag particles on the as-deposited films were irregular, but those on the annealed films became polyhedrons with sharp edges and corners. This 360 °C-annealing increased the SERS enhancement factor (EF) up to 5.12 × 10^8^ and reached a detection limit lower than 10^−10^ M for crystal violet [28]. Because of its low cost, high level of control, and reliability, DC sputtering is favorable for fabricating SERS substrates.

In this study, SERS substrates of single-layered silver nanoparticles (AgNPs) were fabricated by using the DC magnetron sputtering. To attain the reusability of these substrates, TiO_2_ was first deposited on glass slides with the arc ion plating method. As a photocatalyst, TiO_2_ possesses advantages of good physical and chemical stability, strong oxidizing ability, low price, and nontoxicity. In nature, TiO_2_ exists in three crystal phases, namely tetragonal anatase, rutile, and orthorhombic brookite. The anatase structure has the best photocatalytic effect because its surface area is larger than rutile, leading to larger reaction areas for surface water molecules. After being irradiated by ultraviolet light (385 nm), the electrons on the surface of TiO_2_ are released, and positively charged electron holes are formed where the hydroxyl groups (OH-) released by the nearby water molecules are oxidized (i.e., their electrons are captured) and become very active hydroxyl radicals. Moreover, once an organic substance encounters these radicals, its electrons will be captured. Due to the breakdown of its bonds, this substance will disintegrate, and water and carbon dioxide are finally produced. The self-cleaning function based on TiO_2_ photocatalytic degradation enables recyclability. For example, by using a hydrothermal method, a Ag-deposited TiO_2_ flower-like nanomaterial SERS-active substrate was reported. This recyclable substrate exhibited ultrasensitivity with an LOD of 4.47 × 10^–16^ M for Malachite green [29]. Jiang et al. grew AgNPs on the surface of TiO_2_ nanospheres to complete a self-cleaning function. They exhibited an LOD lower than 10^−12^ mol/L for rhodamine 6G (R6G) molecule in a recyclable manner [30]. Other Ag-TiO_2_-based recyclable SERS substrates are reported in references [31,32,33]. In the present work, silver monolayers of different thicknesses were prepared to investigate the resulting Raman spectra. The surface morphologies and cross-sections of these TiO_2_/AgNP substrates were evaluated by using field-emission scanning electron microscopy (FESEM). Their structures were examined by using X-ray diffraction (XRD), X-ray photoelectron spectroscopy (XPS), and energy-dispersive X-ray spectroscopy (EDS) analysis. To test the sensitivity of these substrates, SERS spectra of R6G analytes in different concentrations were collected for comparison. The limit of detection (LOD) and enhancement factor (EF) were 10^−8^ M and 1.01 × 10^9^, respectively. To evaluate the reusability, the same (all R6G) or different (R6G-paraquat-acetylcholine) samples were detected consecutively. With at least five cycles, these substrates exhibited good stability and repeatability. Moreover, they are able to maintain sensitivity after being stored in a vacuum desiccator or a moisture-proof box for up to 40 weeks [34]. With the advantages of low cost, high production rate, high sensitivity, reusability, and long-term stability, the proposed SERS substrates provide a label-free platform for detecting various chemical and biological substances on surfaces and in solutions.

## 2. Results and Discussion

### 2.1. Surface Morphologies of TiO_2_/AgNP Substrates

Usually, to increase the EF of SERS, TiO_2_ films with thicknesses of a few tens of nm are fabricated together with the nanoparticles [35,36]. The mechanisms responsible for such enhancement were reported to be the product of concomitant chemical and electromagnetic effects with an important contribution from plasmon-induced charge transfer [35]. The structure of these TiO_2_ films deposited with the atomic layer deposition (ALD) or sol-gel dip coating methods was confirmed to be amorphous, resulting in inadequacy as the photocatalyst. To prepare TiO_2_ in the desired anatase phase, chemical methods such as the sol-hydrothermal one are commonly used, but with these means, the growth rate cannot be precisely controlled [29,32]. In this study, the TiO_2_ films were prepared by using the arc ion plating method to attain a stable deposition rate of about 4.8 μm/h. This rate is better than those derived from physical vapor deposition (PVD)-based methods. Figure 1a shows the cross-sectional FE-SEM image of the TiO_2_ layer, indicating that the thickness of this layer was about 2 μm. As the deposition time increased, the thickness of the TiO_2_ film increased correspondingly. In response to this, the structure gradually changed from amorphous to columnar crystalline. Silver layers of different thicknesses from 3 to 6 nm were sputtered on TiO_2_-deposited glass slides at a rate of 2.3 nm/min. Without the TiO_2_ layer, the AgNP films were transparent, and their grain sizes were analyzed from the FE-SEM images. For example, these values are about 18.8 and 31.9 nm for thicknesses of 3 and 6 nm, respectively [34]. This resulted in average sizes of hotspots of around 5.2 nm due to the self-assembly of AgNPs during deposition [34]. By leveraging the self-assembly of block copolymer colloids, Rastogi et al. demonstrated gold nanoparticle cluster arrays (NCAs) with intercluster hotspots of 10 nm in size [37]. As shown in Figure 1b, in the presence of the TiO_2_ layer, the boundaries of AgNPs became vague, so it is difficult to measure the sizes of these particles. However, from the following structure analysis and SERS data, it is suggested that AgNPs with grain sizes around 20~30 nm were deposited on TiO_2_-coated glass slides [34].

### 2.2. Structure Analysis of TiO_2_/AgNP Substrates

XRD patterns were used to examine the crystal phase composition of the TiO_2_/AgNP substrate. As indicated in Figure 2, the XRD peaks of Ag were suppressed by those of TiO_2_, as the thickness of the Ag nanoparticles was only 6 nm. The anatase phase was attained by inspecting the characteristic diffraction phases at 2θ = 25.27° (101), 36.88° (103), 37.70° (004), 38.51° (112), and 47.98° (200). With deposition times less than 15 min, the rutile structure was formed due to insufficient temperature. When the time increased beyond 25 min, a strong peak indicating the TiO_2_ anatase (101) plane was clearly observed due to the increase in deposition temperature up to 200 °C. The structure of TiO_2_ also depended on the pressure of O_2_, being unobservable, amorphous, rutile/anatase mixture, and anatase under 1.1 × 10^−3^, 1.9 × 10^−3^, 2.6 × 10^−3^, and 3.8 × 10^−3^ torr, respectively.

To analyze the chemical valence states and composition of the TiO_2_/AgNP substrate, XPS measurements were performed with C 1 s at 284.5 eV selected for calibration. Figure 3a shows the full-range spectrum, indicating the substrate was composed of Ag, Ti, and O elements. Figure 3b illustrates the Ag 3d spectrum of the substrate, showing two peaks centered at 368.2 and 374.2 eV. These two values correspond to the Ag 3d_5/2_ and Ag 3d_3/2_ binding energies, respectively, with the splitting energy of 6 eV. The full widths at half maximum (FWHMs) of the Ag 3d_5/2_ and Ag 3d_3/2_ peaks are 0.649 and 0.643 eV, respectively. Further analysis of the 3d5/2 peak reveals that this peak consists of a significant one of Ag(0) at 368.2 eV (91%) and a fainter one of Ag(I) at 368.7 eV (9%, corresponding to Ag_2_O). After being stored in the vacuum desiccator for 40 weeks, the compositions of Ag(0) and Ag(I) in the 3d_5/2_ peak changed to 78% and 22%, suggesting increased oxidation. The Ti 2p spectrum is shown in Figure 3c, with two peaks at 458.9 and 464.7 eV corresponding to the Ti 2p_3/2_ and Ti 2p_1/2_ binding energies, respectively. This is attributed to the presence of Ti^4+^ in TiO_2_. Small peaks at ~456.5 and ~462 eV for Ti^3+^ and 455.2 and 460.8 eV for Ti^2+^ appear as TiO_2_ suboxides [38]. Finally, Figure 3d indicates the O 1 s spectrum, where a peak centered at 530.1 eV is originated from the Ti-O binding.

The EDS pattern of the TiO_2_/AgNP substrate is shown in Figure 4a, confirming that it was composed of Ag, Ti, and O elements. The percentage composition was 0.8% Ag, 46% Ti, and 53.2% O. Ag was found to be much less than Ti and O simply because the thickness of the AgNP film (6 nm) was much smaller than that of the TiO_2_ layer (2 μm). Figure 4b–d suggests that Ti (green), O (red), and Ag (green) elements were uniformly distributed in the substrate, respectively.

### 2.3. SERS of TiO_2_/AgNP Substrates

The Raman spectrum of the TiO_2_-deposited glass slide is shown in Figure 5a. There are six basic Raman-active modes for the anatase phase with the representation as A_1g_  +  2B_1g_  +  3E_g_ [39]. The spectrum confirms the features of the TiO_2_ anatase with peaks at 144 (E_g_), 196 (E_g_), 398 (B1_g_), 514 (A1_g_), and 638 (E_g_) cm^−^^1^. Rhodamine 6G (R6G), a highly fluorescent dye, is commonly and widely used as a tracer in water to determine the flow rate. R6G molecule is composed of two chromophores, a dibenzopyrene chromophore (xanthene) and a carboxyphenyl group tilted by ~90° with respect to the xanthene ring [40]. When excited with visible light, it exhibits the so-called surface-enhanced resonance Raman scattering (SERRS), the combination of a molecular resonance Raman effect and the SERS effect [40]. Due to such enhancement, R6G interacting with metal particles has been used as the model for verifying the sensitivity of SERS-based biosensors. Raman bands in SERS spectra of R6G include 612 (C–C–C ring in-plane bending), 773 (C–H out-of-plane bending), 1183 (C–H in-plane bending), 1310 (Aromatic C–C stretching), 1360 (Aromatic C–C stretching), 1507 (Aromatic C–C stretching), 1595 (Aromatic C–C stretching), and 1648 (Aromatic C–C stretching) cm^−^^1^ [41]. The SERS spectra resulting from R6G analytes at a concentration of 10^−^^6^ M on the TiO_2_/AgNP substrates with different Ag thicknesses are illustrated in Figure 5b, indicating characteristic peaks of R6G. Signals from the AgNP films of 4.5, 6, and 7.5 nm seemed higher than those of 3 and 9 nm. The 6 nm AgNP film represents the optimal SERS enhancement because all of its peaks were higher than those from other thicknesses. This could also be verified by integrating the spectra of all thicknesses and inspecting the one with a maximum area. Therefore, the TiO_2_/AgNP substrate with the Ag thickness of 6 nm was used in all other experiments.

The performance of the present TiO_2_/AgNP substrates in terms of sensitivity was evaluated by conducting SERS measurements of R6G at titrated concentrations from 10^−5^ to 10^−10^ M. As shown in Figure 6a, the LOD was about 10^−8^ M. The enhancement factor (EF) was calculated at the dominant peak at 612 cm^−1^ by using the equation EF = I_SERS_C_RS_/I_RS_C_SERS_, where I_SERS_ and I_RS_ are the Raman intensities of the analytes adsorbed on the AgNPs substrate at a concentration of C_SERS_ and bare substrate at a concentration of C_RS_, respectively. In Figure 6b, I_SERS_ of 11,000 a.u. (arbitrary unit) was obtained with C_SERS_ of 10^−8^ M, and I_RS_ of 108.6 a.u. for the TiO_2_ substrate was obtained with C_RS_ of 10^−1^ M. These resulted in a SESR EF of 1.01 × 10^9^. To compare the interference of the TiO_2_ peak, which has the same vibrational bands of R6G at 612 cm^−1^, the Raman pattern of the glass slide in our previous study is shown in Figure 6b [34]. Its I_RS_ and EF values are 93.4 a.u. and 1.17 × 10^9^, respectively. The interference can be neglected in the measurement of R6G.

In terms of LOD and EF for R6G, the present SERS substrates, combining simple sputtering technology and cheap glass slides, were better than or at least comparable to others. Zhang et al. reported arrayed nanoporous silver thin films with unique anisotropic morphologies and nanoporous structures on polymethylmethacrylate (PMMA) substrates, exhibiting an R6G LOD of 10^−6^ M and a SERS EF of more than 10^6^ [42]. Another SERS platform employed large-area, hexagonal-wrapped ZnO nanorod arrays incorporating Ag nanoparticles to achieve an R6G LOD of 10^−6^ M and a SERS EF up to 4.2 × 10^7^ [43]. To improve the LOD and EF of semiconductor-based SERS substrates, a strategy involving oxygen incorporation in MoS_2_ was reported, showing an EF up to 100,000-fold and an LOD below 10^−7^ M for R6G compared with oxygen-unincorporated samples [44]. In other studies, LODs of 10^−9^, 10^−10^, and 10^−11^ M were realized in photocatalytic recyclable SERS substrates of Au nanoparticles grown on TiO_2_ nanowire arrays, decorated TiO_2_ and gold nanoparticles on reduced graphene oxide nanosheets, and Fe_3_O_4_ core/TiO_2_ interlayer/Au shell, respectively [32].

### 2.4. Uniformity, Reproducibility, and Recyclability Tests of TiO_2_/AgNP Substrates

To test the uniformity (different spots of the same substrate) and reproducibility (different substrates) of the present TiO_2_/AgNP substrates, 10^−6^ M R6G solution was dropped onto three random spots of each of the three random substrates. The three-dimensional (3D) spectra from nine Raman signals are shown in Figure 7a. As illustrated in Figure 7b, these intensities at 612 cm^−1^ exhibited a relative standard deviation (RSD) of 2.19%, which indicates excellent uniformity and reproducibility. Moreover, after being stored in a vacuum desiccator or a moisture-proof box for 40 weeks, the substrates remained good quality with RSDs in R6G Raman intensities of 3.3~5% over time [34]. With the screen printing method, Wu et al. reported large-area SERS substrates composed of Ag nanoparticles on plastic polyethylene terephthalate (PET) [45]. As shown, the SERS detection limit of R6G was higher than 10^−10^ M, and the RSD value for 784 spots on the substrate was less than 20% [45]. By using the roll-to-plate embossing technology and a hydrothermal method, SERS substrates based on an ordered micropyramid array and silver nanoparticles were fabricated. The minimum RSD for the substrate was calculated as 4.99%, and SERS performance basically had no loss after 12 days of placement [46]. Another SERS substrate consisting of wafer-scale uniformly hydrophobic silicon nanorods arrays decorated with Au nanoparticles exhibited RSDs of 4.04~6.14% over the intensities of R6G Raman spectra from 16 random positions of the substrate [47]. A SERS substrate based on a ternary film-packaged, silver-coated gold-nanoparticle plasmonic array demonstrated long-term storage stability of up to two months at room temperatures without significant changes in SERS signals [48].

The self-cleaning and recycling abilities of the present TiO_2_/AgNP substrates were verified by conducting photocatalytic degradation involving TiO_2_. This reaction concerning R6G is [49]:$$C_{28}H_{31}N_{2}O_{3}\mathrm{Cl}\overset{\frac{{\mathrm{TiO}}_{2}}{\mathrm{Ag}}+\;h\nu }{\to }{\mathrm{CO}}_{2}+H_{2}O+{\mathrm{HNO}}_{3}+\mathrm{HCl}\ldots \left(\mathrm{or}\;\mathrm{other}\;\mathrm{small}\;\mathrm{molecules}\right)$$

The Raman spectra of 10^−6^ M R6G solution after UV irradiation at different time points from 0 to 120 min are illustrated in Figure 8a. As indicated, the signals (especially the peak at 612 cm^−1^) were very weak after 90 min of UV irradiation and then became almost invisible at 120 min. Figure 8b shows the 612 cm^−1^ Raman intensities at different time points of UV exposure for both AgNP-only and TiO_2_/AgNP substrates. These intensities remained almost the same without the presence of TiO_2_, while they decreased at a certain rate due to the photocatalytic reaction of R6G. The photocatalytic rate (k) could be calculated by fitting the data to the equation $I=I_{0}e^{-kt}$, where *I*_0_ and *I* are the intensities at time = 0 and t, respectively. The calculated rate is about 0.029 min^−1^ at the R6G Raman peak of 612 cm^−1^. Zhang et al. evaluated the SERS and photocatalytic performance of three different Ag/TiO_2_/graphene(G) composites [50]. The calculated EFs for Ag-G-TiO_2_, G-Ag-TiO_2_, Ag-TiO_2_, and Ag-TiO_2_-G substrates were 1.1 × 10^6^, 5.4 × 10^5^, 7 × 10^5^, and 5 × 10^5^, respectively. Moreover, the calculated photocatalytic rates were about 0.0371, 0.0301, and 0.0111 min^−1^ for Ag-G-TiO_2_, G-Ag-TiO_2_, and Ag-TiO_2_-G substrates, respectively. A much higher R6G degradation reaction rate of 0.05764 min^−1^ was reported on the SERS substrate with deposited silver nanoparticles on a titania nanopore array [51]. Besides TiO_2_, other metal oxides can serve as photocatalysts in recyclable SERS substrates. For example, an Ag/polydopamine/ZnO SERS substrate was fabricated to exhibit a photocatalytic rate of 0.0391 min^−1^ [52]. Other photocatalysts such as WO_3_, Fe_3_O_4_, Cu_2_O, Ta_2_O_5_, Nb_2_O_5_, and MoO_2_ have also been incorporated into SERS substrates to attain recyclability [53]. It is concluded that the photocatalytic rate depends not only on the type of photocatalysts but also on the structure and how it is integrated into the SERS substrate.

Five cycles of sample-dropping, spectrum-recording, self-cleaning, and spectrum-recording were performed to test the reusability of the present TiO_2_/AgNP substrates. The self-cleaning process was carried out with a UV exposure of 120 min. Figure 9a shows the Raman spectra of 10^−6^ M R6G solution in five repeats. The intensities at 612 cm^−1^ were plotted versus recyclable times in Figure 9b. In the presence of R6G, the intensities decreased gradually from 56,738 to 36,909, 36,745, 31,520, and 28,779 after the first, second, third, and fourth photocatalytic degradations, respectively. Although the intensity dropped to about 50% of the original value after four times of UV irradiation and degradation, the signal was still strong enough for conducting further measurements. The number of reused times presented here was comparable to others [32,54], and using UV provided an easy and noncontact degradation method without the need of exposing the substrate to certain chemicals such as H_2_O_2_ [55]. These results ensure the recyclability, reusability, and applicability of the present SERS substrates.

The present substrate was further used to detect other analytes in a recyclable manner. Figure 10 shows the Raman spectra of 10^−6^ M R6G, 10^−6^ M paraquat, and 10^−6^ M acetylcholine solutions with photocatalytic degradation by a 120 min UV irradiation in between. Paraquat (1,1′-dimethyl-4,4′-bipyridinium dichloride, (C_6_H_7_N)_2_Cl_2_) is one of the most widely used herbicides for chemical weed control. Despite being banned in several countries, it is still popular in many developing countries due to its cheapness and effectiveness. This chemical is considered toxic and even lethal to humans and other animals because its exposure leads to the production of reactive oxygen species (ROS) through a process of redox cycling [56]. Paraquat-induced toxicity has also been linked to the development of Parkinson’s disease [57]. Therefore, it is important to monitor the presence and, if any, the concentration of paraquat in all kinds of water, soils, and foods. As indicated in the figure, Raman peaks of paraquat at 656, 840, 1187, 1300, 1525, and 1635 cm^−1^ were clearly observed. Other SERS substrates were also fabricated for the detection of paraquat [58,59]. Acetylcholine (ACh, CH_3_OCOCH_2_CH_2_N^+^-(CH_3_)_3_), the first discovered neurotransmitter, acts in the central and peripheral nervous system by binding to a variety of cellular targets. It affects many body functions, such as the activation of skeletal muscles, plasticity, arousal and reward in the central nervous system, and even learning and memory. This neurotransmitter is highly associated with Alzheimer’s disease [58], Parkinson’s disease [60], Huntington’s disease [61], and attention-deficit hyperactivity disorder (ADHD) [62]. Being able to detect trace amounts of ACh is hence crucial in understanding its roles in these neurological disorders. For example, Lee et al. reported the spread spectrum SERS (ss-SERS) detection of ACh at the attomolar level by encoding excited light and decoding SERS signals with peak autocorrelation and near-zero cross-correlation [63]. The Raman spectra of ACh obtained with the present TiO_2_/AgNP substrates showed feature signals at 646, 719, 817, 869, 956, 1133, 1215, 1280, 1335, 1446, and 1737 cm^−1^. The LODs and EFs of the present substrates in detecting paraquat and acetylcholine need to be further investigated.

## 3. Materials and Methods

To prepare SERS substrates, glass slides of 76.2 × 25.4 × 1 mm were first polished with wet cotton, moistened with CeO_2_ powders to increase the adhesion of the deposited layers, cleaned ultrasonically for 20 min, and then blown with clean nitrogen gas. The TiO_2_ films were prepared by using the arc ion plating method shown in Figure 11a. This coating system is divided into three parts: the arc power supply system, the vacuum chamber, and the gas exhaust/delivery system. Via O_2_ supply, TiO_2_ compounds were grown on glass slides using reactive coating technology [64]. The experimental parameters are as follows: purity of the Ti target: 99.99%; arc power supply voltage: 20 V; arc power supply current: 75 A; DC bias supply voltage: 75 V; working pressure: 5 × 10^−3^ torr (Ar: O_2_ = 2: 7); Ar flow rate: 100 sccm; O_2_ flow rate: 350 sccm; deposition time: 25 min. Afterward, as shown in Figure 11b, a TiO_2_-deposited glass slide was placed on top of the substrate platform within the vacuum chamber. A silver target of 5 cm in diameter and 99.99% purity was mounted about 150 mm above the substrate on the sputtering gun. To determine the thickness of the sputtered film, a quartz crystal monitoring head (STM-100, Sycon Instrument, East Syracuse, New York, NY, USA) sat next to the substrate. Before sputtering, the chamber was evacuated to a base pressure of 10^−5^ torr with a diffusion pump (VHS-6, Agilent, CA, USA), and then the working gas, argon gas (99.995%), was fed into the chamber at a flow rate of 10 sccm to attain a working pressure of 3 × 10^−3^ torr. The voltage and current of the DC generator (PFG 1500 DC, HÜTTINGER Elektronik, Freiburg, German) were set at 316 V and 0.01 A, respectively. The substrate platform rotated at a speed of 30 rpm to maintain uniform thickness across the surface. A stable deposition rate of 2.3 nm/min was achieved by calibrating with the quartz crystal monitor. Moreover, silver layers of different thicknesses could be obtained by varying the deposition time [34].

After fabrication, these TiO_2_/AgNP substrates were stored in a vacuum desiccator or a moisture-proof box to prevent humidity intrusion. Their surface morphologies and cross-sections were examined with a FESEM (JSM-7600F, Jeol, Tokyo, Japan). EDS analysis was also performed with this instrument. An X-ray diffractometer (D8 Advance, Bruker, MA, USA) with Cu K-α radiation (λ = 0.15406 nm) was used to characterize their structures. An X-ray photoelectron spectroscopy (XPS) (PHI 5000 VersaProbe III, ULVAC-PHI, Kanagawa, Japan) was applied to analyze the chemical bonding states at the surfaces.

To detect Raman signals, analytes of desired concentrations were dropped onto the TiO_2_/AgNP substrates. A Raman spectroscopy (HR Evolution, Horiba, Kyoto, Japan) with a 0.1 mW He-Ne laser (λ = 632.8 nm) light source, focused by a 100× objective lens, irradiated a SERS substrate to record the Raman spectrum from 400–2000 cm^−1^ within 5 s 3 times. First, the spectra of R6G solutions in a titration from 10^−5^ to 10^−10^ M were obtained to figure the LOD. To test the uniformity and reproducibility of these substrates, R6G solutions at a concentration of 10^−6^ M were dropped onto three random positions of three separate substrates. The reusability of these substrates was verified by conducting cycles of sample-dropping, spectrum-recording, and self-cleaning. The TiO_2_-assisted photocatalytic degradation of analytes was performed by irradiating the substrate with a UV lamp (wavelength = 385 nm) for 120 min at room temperature. The experimental processes are illustrated in Figure 12.

## 4. Conclusions

In this study, a recyclable SERS substrate composed of TiO_2_ and AgNPs was successfully fabricated. After fabrication, the surface morphology and cross-section of the substrate were evaluated by using the FESEM, and its structure was examined with XRD, XPS, and EDS analysis. The anatase-phased TiO_2_ was observed to ensure its photocatalytic ability. SERS spectra of R6G in titrated concentrations were measured to obtain LOD of 10^−8^ M and SERS EF of 1.01 × 10^9^. The uniformity and reproducibility of the present substrate was tested on nine spots and showed an RSD of 2.19%. The self-cleaning and recycling ability was verified by conducting photocatalytic degradation via UV irradiation for 120 min. Then, to evaluate its reusability, the same (all R6G) or different (R6G-paraquat-acetylcholine) samples were detected consecutively, indicating the substrate exhibited good stability and repeatability after at least five cycles. Accordingly, these results suggest that the present SERS substrate can serve as a platform for label-free detection of various chemical and biological substances.

## Figures and Tables

**Figure 1 molecules-27-06755-f001:**
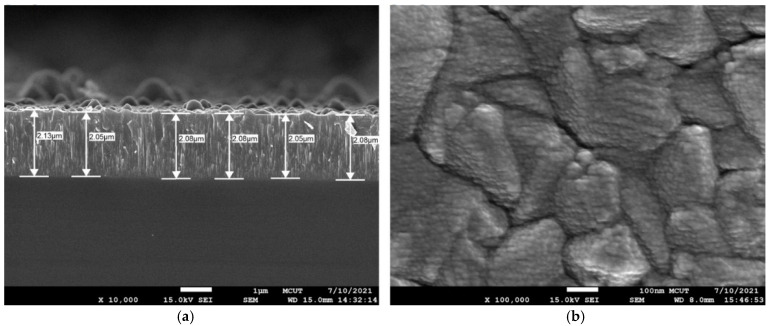
(**a**) Cross-sectional SEM image of the TiO_2_ layer. Scale bar = 1 μm. (**b**) SEM image of the 6 nm thick Ag film on the TiO_2_-deposited glass slide. Scale bar = 100 nm.

**Figure 2 molecules-27-06755-f002:**
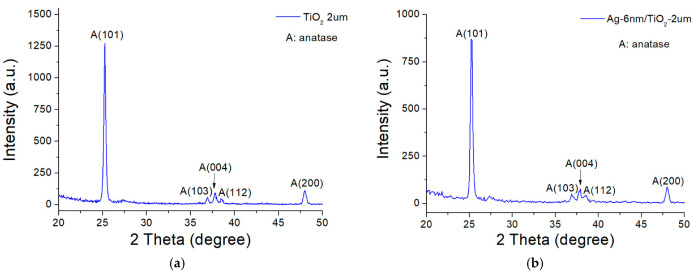
XRD patterns of (**a**) TiO_2_/substrate and (**b**) AgNPs of 6 nm thickness/TiO_2_/substrate.

**Figure 3 molecules-27-06755-f003:**
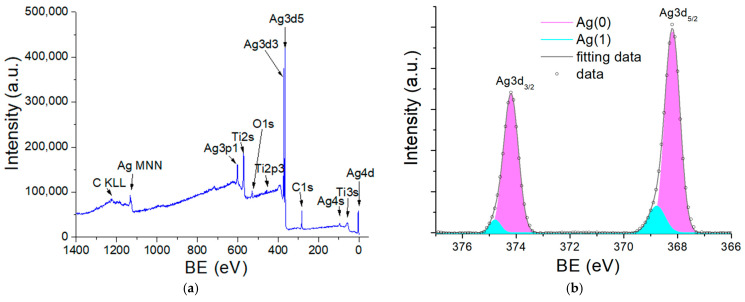
XPS scanning of the TiO_2_/AgNP substrate in different regions: (**a**) Full spectrum, (**b**) Ag 3d region, (**c**) Ti 2p region, (**d**) O 1s region.

**Figure 4 molecules-27-06755-f004:**
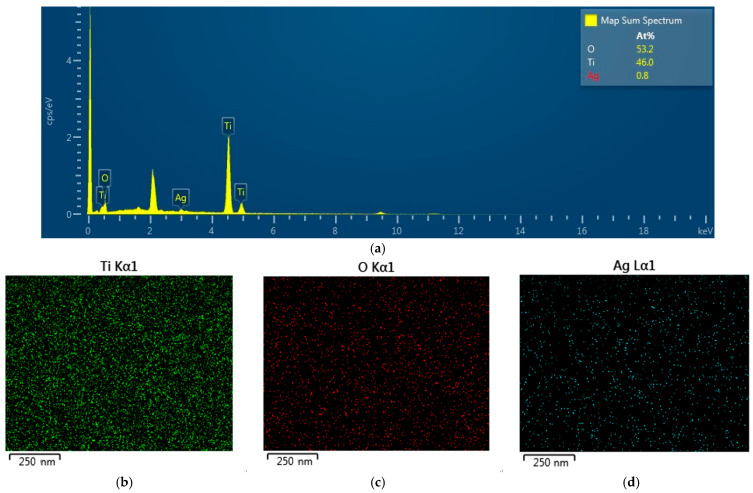
EDS analysis of the TiO_2_/AgNP substrate: (**a**) EDS spectrum, (**b**–**d**) elemental mapping images including elements Ti, O, and Ag.

**Figure 5 molecules-27-06755-f005:**
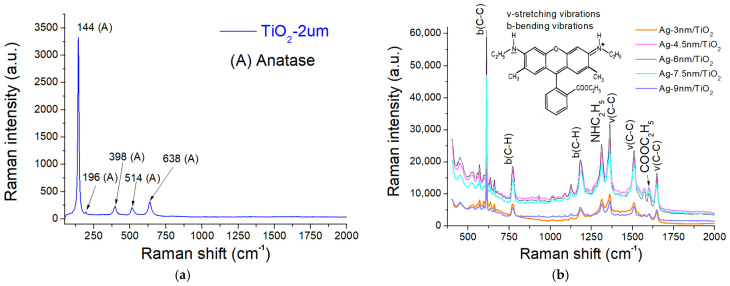
(**a**) Raman spectrum of the TiO_2_-deposited glass slide. (**b**) SERS spectra resulting from R6G solution (10^−6^ M) on TiO_2_/AgNP substrates with different Ag thicknesses.

**Figure 6 molecules-27-06755-f006:**
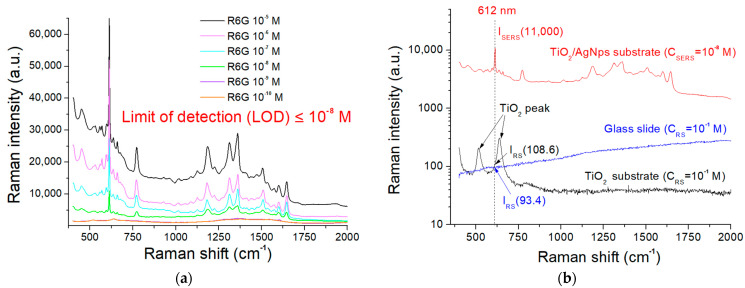
(**a**) SERS signals of various concentrations of R6G. (**b**) SERS signal of TiO_2_/AgNP substrate and Raman signals of bare TiO_2_ substrate and glass slide.

**Figure 7 molecules-27-06755-f007:**
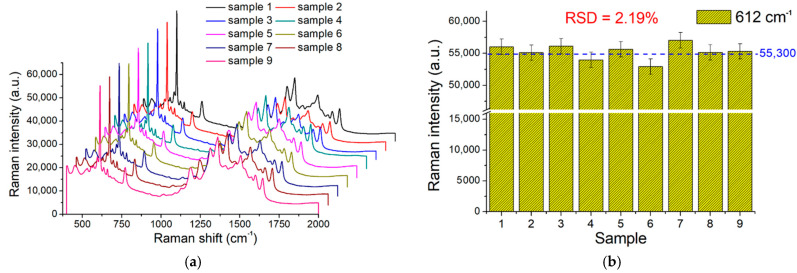
(**a**) 3D Raman spectra of 10^−6^ M R6G solution from nine random spots on the substrate. (**b**) Intensities of the spectra in (**a**) at 612 cm^−1^.

**Figure 8 molecules-27-06755-f008:**
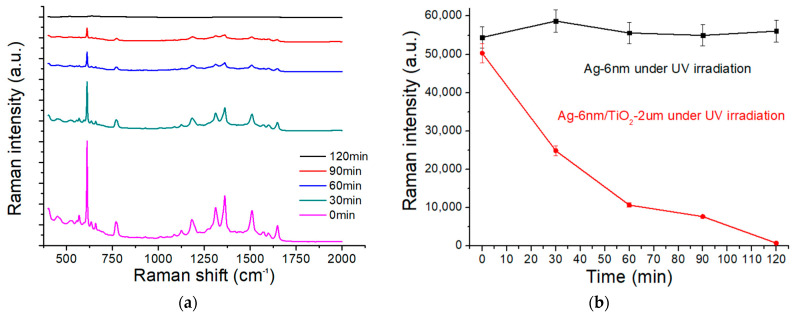
(**a**) Raman spectra of 10^−6^ M R6G solution after UV irradiation at different time points. (**b**) Raman intensities of the spectra in (**a**) at 612 cm^−1^ after UV exposure at different time points for both AgNP-only and TiO_2_/AgNP substrates.

**Figure 9 molecules-27-06755-f009:**
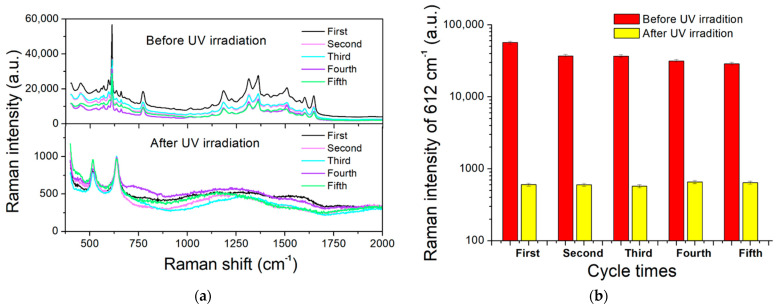
(**a**) Raman spectra of 10^−6^ M R6G solution in five repeats. (**b**) Intensities at 612 cm^−1^ versus recyclable times.

**Figure 10 molecules-27-06755-f010:**
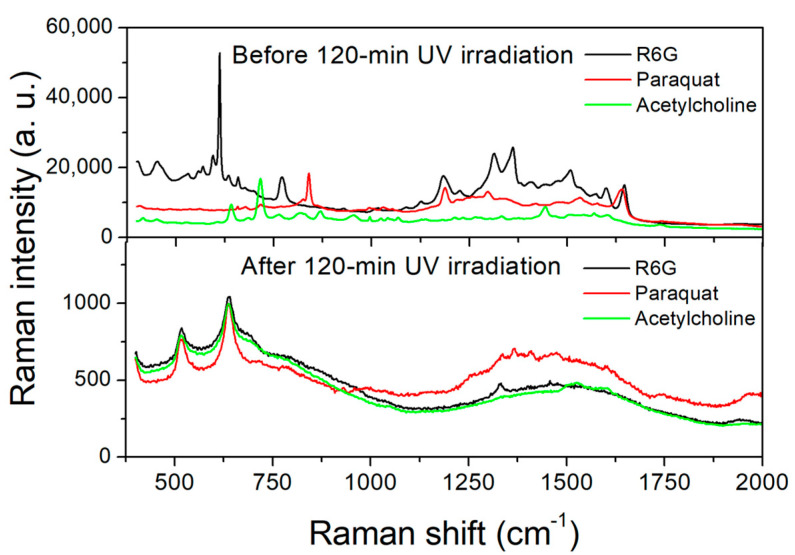
Raman spectra of 10^−6^ M R6G, 10^−6^ M paraquat, and 10^−6^ M acetylcholine solutions with photocatalytic degradation in between.

**Figure 11 molecules-27-06755-f011:**
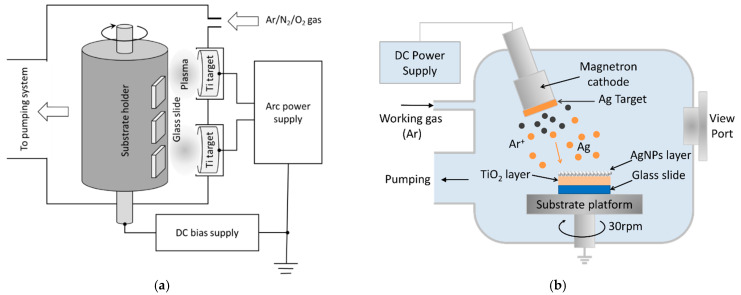
Schematics of (**a**) the arc ion plating system and (**b**) the DC sputtering system.

**Figure 12 molecules-27-06755-f012:**
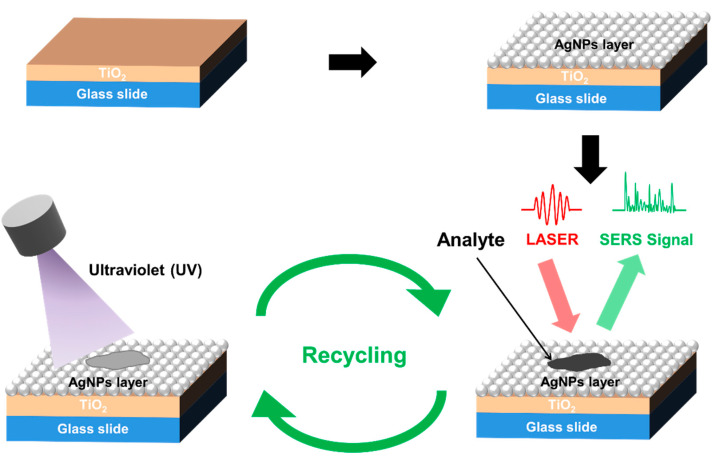
Experimental processes.

## Data Availability

The data are included in the article.

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
