# Peer review of "Highly Sensitive, Robust, and Recyclable TiO2/AgNP Substrate for SERS Detection"

_molecules, 2022, doi:10.3390/molecules27196755_

Round 1

Reviewer 1 Report

The manuscript entitled High sensitive, cost-effective, robust, and recyclable TiO2/AgNPs substrate for SERS label-free detection represents and interesting research on preparation of recyclable TiO2/Ag nanoparticle (AgNP) substrates by using simple arc ion plating and direct-current (dc) magnetron sputtering technologies. Self-cleaning was performed via UV irradiation, and recyclability was achieved after at least five cycles of detection and degradation.

Introduction part should be enriched with recent references

Conclusion part should be brief and informative, not with too many fabrication details. Conclusion should contain only the main conclusion of the manuscript.

The Authors should include more coments on photocatalytic rate of the plate.

Did the Authors perform the measurements to detect potential inclusion in the structure of the plate?

The results were well presented and discussion should be extended according to the comments.

Author Response

Response to Comments and Suggestions

Reviewer 1

Comments and Suggestions for Authors

The manuscript entitled High sensitive, cost-effective, robust, and recyclable TiO2/AgNPs substrate for SERS label-free detection represents and interesting research on preparation of recyclable TiO2/Ag nanoparticle (AgNP) substrates by using simple arc ion plating and direct-current (dc) magnetron sputtering technologies. Self-cleaning was performed via UV irradiation, and recyclability was achieved after at least five cycles of detection and degradation.

Response:

The authors thank this reviewer for all helpful comments.

Introduction part should be enriched with recent references

Response:

We have modified the Introduction part accordingly to include few more recent references.

Conclusion part should be brief and informative, not with too many fabrication details. Conclusion should contain only the main conclusion of the manuscript.

Response:

We have modified the Conclusion part accordingly to eliminate fabrication details.

The Authors should include more coments on photocatalytic rate of the plate.

Response:

We did include quite a few comments on the photocatalytic rate indicating how this value depends on (1) the type of photocatalysts and (2) its structure and how it is integrated into the SERS substrate. Please refer to Section 3.4.

Did the Authors perform the measurements to detect potential inclusion in the structure of the plate?

Response:

As described in Section 3.2, XRD patterns were used to examine the crystal phase composition of the TiO2/AgNPs substrate. Also, XPS measurements were performed to analyze the chemical valence states and composition of the substrate. Finally, the EDS pattern of the substrate confirmed that it was composed of Ag, Ti, and O elements.

The results were well presented and discussion should be extended according to the comments.

Response:

We have modified the manuscript accordingly.

Reviewer 2 Report

In this work, the authors fabricate TiO2/Ag composite SERS substrates through arc ion plating and direct current (DC) magnetron sputtering technologies and evaluate their SERS performance. It could be interesting, but I prefer to suggest a major modification before publication due to numerous questions. I hope I can receive the revised manuscript for further evaluation. My suggestions and comments are as follows.

  1. The authors emphasize “cost-effective” in the title. However, because the TiO2/Ag substrate is made with metal targets, the TiO2 layer is as thick as 2 um, and sputtering methods could suffer from low utilization due to the waste on the chamber walls, the cost of this presented substrate could not be lower than the pure Ag NPs substrate. Compared with many reported substrates, this could not be competitive in this aspect.
  2. The authors use “robust” to describe the SERS substrate in the title. Does it mean the substrate can be reused, if so, why the author also emphasizes “recyclable” in the title? It just repeats. In a general understanding, a robust substrate refers to the substrate that can be stable against something like an acidic environment and the air, but I cannot find any discussion about this in the manuscript.
  3. This is about “label-free” in the title. Most SERS analyses are label-free, why it should be emphasized here? Additionally, the experiments were performed by dropping the probe molecule solution on the substrate for detection, and it is no means to discuss whether the detection is label-free or not.
  4. I suggest the introduction part needs to be reorganized. It should focus on the theme of SERS. Too much inconsequential introduction about the biosensors and analytical devices is presented.
  5. On line 89, page 2, the authors state that SERS is selectivity, but it seems not general merit of the common SERS. A selectively SERS substrate should be highly designed.
  6. This manuscript should improve on the organization. For example, in section 3.1 Surface morphologies of TiO2/AgNPs substrates. This part should introduce the main results of the authors’ findings; however, the main contents focus on the background introduction. Some similarities can be found somewhere. Please check them.
  7. Figure 2 gives the experimental processes. My question is why the AgNP layer on the substrate changes in morphology. In addition, to avoid misunderstandings, I suggest changing the “substrate” name of the glass layer in Figures 1 and 2.
  8. On lines 227 to 229, page 6, the authors state the Ag NPs on the substrate are about 20 ~ 30 nm according to the structural analysis and SERS data, but no detailed discussion about this can be found.
  9. More details about the SERS measurement should be disclosed, like the exciting wavelength, integral time, substrate size, etc.
  10. On line 297, page 9, the “limit of detection (LOD)” can be replaced by LOD since it has been defined in the introduction part.
  11. In Figure .5c, the Ti 2p XPS peak is shown. Some other peaks in the spectrum should be identified or explained since they are much stronger than the noise level.
  12. The caption of Figure 7 should be changed because the Raman spectra of the TiO2-deposited glass slide (A) cannot be called SERS.
  13. The SERS pattern of the blank TiO2/AgNPs substrates should be provided in Figure 7 or somewhere.
  14. For the SERS enhancement factor (EF), the SERS and normal Raman of the same vibrational bands of R6G need to measure. However, in the data shown in Figure 8b, the normal Raman pattern does not show any R6G vibration bands. The so-called IRS (108.6) seems greatly contributed by the TiO2 band. In addition, the authors also confuse Raman with SERS in the caption of Figure 8. To avoid the interference of the TiO2 band, I suggest using a glass slide for the normal Raman measurement.
  15. In Figure 8a, I cannot get the meaning of the red text description of “limit of detection (LOD) =>10-8 M”. It could be: LOD≤10-8 M.
  16. On lines 301-302, the “ISERS of 11003” and “IRS of 108.6” should have units.
  17. The authors have an ambiguous demonstration of reproducibility. The data achieved from different substrates can show reproducibility. Signals from the same substrate can show uniformity. Thus, I am willing to suggest the authors study the uniformity first and put the data in Figure 7 or somewhere; based on this, the reproducibility (signals from different substrates) can be then studied.
  18. It is much better to make the expressions of “TiO2/Ag nanoparticle (AgNP) substrates” and “TiO2/Ag AgNPs substrates” consistent in the whole content.
  19. As a highly refractive and catalysis-active material, the TiO2 layer thickness should have some influence on the detection and recyclability. It is much better to provide the relevant data.
  20. The actual problem may be far more than that I have pointed. The author should try to check and modify it by himself.

Author Response

Response to Comments and Suggestions

Comments and Suggestions for Authors

In this work, the authors fabricate TiO2/Ag composite SERS substrates through arc ion plating and direct current (DC) magnetron sputtering technologies and evaluate their SERS performance. It could be interesting, but I prefer to suggest a major modification before publication due to numerous questions. I hope I can receive the revised manuscript for further evaluation. My suggestions and comments are as follows.

Response:

The authors thank this reviewer for all helpful comments.

  1. The authors emphasize “cost-effective” in the title. However, because the TiO2/Ag substrate is made with metal targets, the TiO2 layer is as thick as 2 um, and sputtering methods could suffer from low utilization due to the waste on the chamber walls, the cost of this presented substrate could not be lower than the pure Ag NPs substrate. Compared with many reported substrates, this could not be competitive in this aspect.

Response:

We agreed with this point of view that the present substrate is not as cost-effective as others. We have modified the title to avoid any ambiguity.

  1. The authors use “robust” to describe the SERS substrate in the title. Does it mean the substrate can be reused, if so, why the author also emphasizes “recyclable” in the title? It just repeats. In a general understanding, a robust substrate refers to the substrate that can be stable against something like an acidic environment and the air, but I cannot find any discussion about this in the manuscript.

Response:

The term “robust” indicates that this substrate can be reused for many times without losing its function. As stated in Section 3.4, five repeats were performed to test the reusability of the present substrate. The term“recyclable” simply suggests that it can be reused at least once. These two terms have different definitions in describing SERS substrates. Please see the following references:

  1. B Bassi et al 2019 Nanotechnology 30 025302.
  2. Journal of Colloid and Interface Science Volume 418, 15 March 2014, Pages 87-94.
  3. Nanotechnology 2019 Jan 11;30(2):025302. doi: 10.1088/1361-6528/aae9b3.
  4. Journal of Colloid and Interface Science Volume 418, 15 March 2014, Pages 87-94.

  1. This is about “label-free” in the title. Most SERS analyses are label-free, why it should be emphasized here? Additionally, the experiments were performed by dropping the probe molecule solution on the substrate for detection, and it is no means to discuss whether the detection is label-free or not.

Response:

We agreed with the first part of this comment that SERS analyses are label-free. Therefore, we have modified the title to avoid any ambiguity. However, the probes that are dropped on the substrate can be labeled to detect fluorescent signals as in the case of typical microarray detections.

  1. I suggest the introduction part needs to be reorganized. It should focus on the theme of SERS. Too much inconsequential introduction about the biosensors and analytical devices is presented.

Response:

We have shortened the Introduction part to focus more on SERS.

  1. On line 89, page 2, the authors state that SERS is selectivity, but it seems not general merit of the common SERS. A selectively SERS substrate should be highly designed.

Response:

We agreed with this point of view that a selectively SERS substrate should be specially designed. We have deleted this term to avoid any ambiguity.

  1. This manuscript should improve on the organization. For example, in section 3.1 Surface morphologies of TiO2/AgNPs substrates. This part should introduce the main results of the authors’ findings; however, the main contents focus on the background introduction. Some similarities can be found somewhere. Please check them.

Response:

We have modified the manuscript accordingly. Since Section 3 should include both Results and Discussion, we did explain why a thicker TiO2 film is required.

  1. Figure 2 gives the experimental processes. My question is why the AgNP layer on the substrate changes in morphology. In addition, to avoid misunderstandings, I suggest changing the “substrate” name of the glass layer in Figures 1 and 2.

Response:

We have modified these two figures accordingly.

  1. On lines 227 to 229, page 6, the authors state the Ag NPs on the substrate are about 20 ~ 30 nm according to the structural analysis and SERS data, but no detailed discussion about this can be found.

Response:

This was found and reported in our previous paper. We have added a reference accordingly.

  1. More details about the SERS measurement should be disclosed, like the exciting wavelength, integral time, substrate size, etc.

Response:

We have added more details about the SERS measurement in the revised manuscript. Line215:  “A Raman spectroscopy (HR Evolution, Horiba, Kyoto, Japan) used a 0.1 mW He-Ne laser (λ = 632.8 nm) light source, focused through a 100x objective lens, and irradiated a SERS substrate to measure the Raman spectrum from 400-2000 cm-1 within 5 seconds 3 times.

The substrate size of glass slides is 76.2 × 25.4 × 1 mm, mentioned in Line 177.

  1. On line 297, page 9, the “limit of detection (LOD)” can be replaced by LOD since it has been defined in the introduction part.

Response:

We have modified the manuscript accordingly.

  1. In Figure .5c, the Ti 2p XPS peak is shown. Some other peaks in the spectrum should be identified or explained since they are much stronger than the noise level.

Response:

We have identified these peaks in the revised manuscript at Line 358. “Small peaks at ~456.5 and ~462 eV for Ti3+ and 455.2 and 460.8 eV for Ti2+ appear as TiO2 sub-oxides” and add a reference [39] : Hsu, J.C.; Lee, C.C.; Chen, H.L.; Kuo, C.C.; Wang, P.W. Investiga-tion of thin TiO2 films cosputtered with Si species. Appl. Surf. Sci. 2009., 255, 4852-4858.

  1. The caption of Figure 7 should be changed because the Raman spectra of the TiO2-deposited glass slide (A) cannot be called SERS.

Response:

We have modified this figure caption accordingly.

  1. The SERS pattern of the blank TiO2/AgNPs substrates should be provided in Figure 7 or somewhere.

Data?

Response:

The SERS pattern of the blank TiO2/AgNPs substrates can be seen in Figure 11(a), which is like the Raman spectrum after UV irradiation.

  1. For the SERS enhancement factor (EF), the SERS and normal Raman of the same vibrational bands of R6G need to measure. However, in the data shown in Figure 8b, the normal Raman pattern does not show any R6G vibration bands. The so-called IRS (108.6) seems greatly contributed by the TiO2 band. In addition, the authors also confuse Raman with SERS in the caption of Figure 8. To avoid the interference of the TiO2 band, I suggest using a glass slide for the normal Raman measurement.

Response:

We have modified the caption of Figure 8 accordingly. To evaluate the SERS enhancement factor (EF), the effect of AgNPs instead of TiO2/AgNPs should be investigated. Therefore, it would be fairer to use the SERS signal of R6G from the TiO2/AgNPs substrate as ISERS and the Raman signal of R6G from the TiO2 substrate (not from bare glass slide) as IRS. Please see the following references:

  1. ACS Omega 2021, 6, 41, 27271–27278 (Figure 2)
  2. Journal of Food Measurement and Characterization volume 16, pages 2890–2898 (2022) (Figure 3)

  1. In Figure 8a, I cannot get the meaning of the red text description of “limit of detection (LOD) =>10-8 M”. It could be: LOD≤10-8 M.

Response:

We have modified this in the revised manuscript.

  1. On lines 301-302, the “ISERS of 11003” and “IRS of 108.6” should have units.

Response:

They are all in an arbitrary unit (a. u.) as labeled in the figure. We have modified the manuscript accordingly.

  1. The authors have an ambiguous demonstration of reproducibility. The data achieved from different substrates can show reproducibility. Signals from the same substrate can show uniformity. Thus, I am willing to suggest the authors study the uniformity first and put the data in Figure 7 or somewhere; based on this, the reproducibility (signals from different substrates) can be then studied.

Response:

To avoid a lengthy manuscript and too many figures, we described the combined test on the uniformity and reproducibility of the substrate in the revision.

  1. It is much better to make the expressions of “TiO2/Ag nanoparticle (AgNP) substrates” and “TiO2/Ag AgNPs substrates” consistent in the whole content.

Response:

We have thoroughly checked and modified the manuscript to avoid this confusion.

  1. As a highly refractive and catalysis-active material, the TiO2 layer thickness should have some influence on the detection and recyclability. It is much better to provide the relevant data.

Response:

In the current stage, we don’t have data showing how the thickness of TiO2 layer affects the SERS detection and recyclability. We will try to investigate this in the very near future

  1. The actual problem may be far more than that I have pointed. The author should try to check and modify it by himself.

Response:

We have thoroughly checked and modified the manuscript.

Reviewer 3 Report

In this paper, recyclable TiO2/Ag nanoparticle (AgNP) substrates were prepared using simple arc ion plating and direct- 27 current (dc) magnetron sputtering technologies. The Raman property was investigated by detecting R6G. However, the SERS performance of TiO2/Ag nanoparticle substrates for R6G detection has lower detection limit than the previous reports (Talanta, 210, 2020, 120631; Talanta146, 2016, 533-539),which are shown to be easier to synthesize. Furthermore, There are many questions in the manuscript. Based on these viewpoint, I cannot recommend this manuscript for publication.

1, the manuscript seems to gab for more contents.The Introduction should be concise and just focused on the topic. For example, the 1st and 2nd paragraphs can be shorten or deleted.

In 3.1. Surface morphologies of TiO2/AgNPs substrates, paragraph 1 also introduce too many contents regarding photocatalysis.

2, in Material section, the author retreated the glass substrate, “To prepare SERS substrates, glass slides of 76.2 × 25.4 × 1 mm were first polished with 135 wet cotton, moistened with CeO2 powders,”  regarding “moistened with CeO2 powders”, can authors kindly explain the theory?

3,For the Raman results, the authors should give the detail test conditions. What was laser wavelength? The collection time, scanning times?

4, the function of EF is not correct. the function in the manuscript is only suit for the Raman test in solution.

5. All pictures are low quality.

Author Response

Response to Comments and Suggestions

Comments and Suggestions for Authors

In this paper, recyclable TiO2/Ag nanoparticle (AgNP) substrates were prepared using simple arc ion plating and direct- 27 current (dc) magnetron sputtering technologies. The Raman property was investigated by detecting R6G. However, the SERS performance of TiO2/Ag nanoparticle substrates for R6G detection has lower detection limit than the previous reports (Talanta, 210, 2020, 120631; Talanta146, 2016, 533-539),which are shown to be easier to synthesize. Furthermore, There are many questions in the manuscript. Based on these viewpoint, I cannot recommend this manuscript for publication.

Response:

The authors thank this reviewer for all helpful comments.

The applications of the present SERS substrate are not limited to only “detecting R6G”. Moreover, the “detection limit” is not the only criteria to evaluate the performance of a SERS substrate. Compared with these two reference, the present substrate possesses the following advantages:

  1. It is recyclable and can be reused.
  2. It can be applied to detecting a variety of analytes including paraquat and acetylcholine solutions.
  3. The TiO2 films prepared using the arc ion plating method exhibited the desired anatase phase having the strongest photocatalytic effect.

1, the manuscript seems to gab for more contents. The Introduction should be concise and just focused on the topic. For example, the 1st and 2nd paragraphs can be shorten or deleted.

Response:

We have shortened the Introduction part to focus more on SERS.

In 3.1. Surface morphologies of TiO2/AgNPs substrates, paragraph 1 also introduce too many contents regarding photocatalysis.

Response:

We have shortened this section to avoid introducing too many contents about photocatalysis. However, since it should include both Results and Discussion, we did explain why a thicker TiO2 film is required.

2, in Material section, the author retreated the glass substrate, “To prepare SERS substrates, glass slides of 76.2 × 25.4 × 1 mm were first polished with 135 wet cotton, moistened with CeO2 powders,”  regarding “moistened with CeO2 powders”, can authors kindly explain the theory?

Why?

Response:

In general, a few oxide materials will be grown on the glass when stored for some time. Therefore, we used the wet cotton and moistened it with the CeO2 powders, used in the general optical polish fabrication process, to slightly remove the oxides on the surface to refresh the surface. That process will increase the adhesion of the deposited layers.

Therefore, we revised the text (Line 178) “…. moistened with CeO2 powders to increase the adhesion of the deposited layers, ,…”

3,For the Raman results, the authors should give the detail test conditions. What was laser wavelength? The collection time, scanning times?

Response:

We have added more details about the SERS measurement in the revised manuscript.

Line 215:  “A Raman spectroscopy (HR Evolution, Horiba, Kyoto, Japan) used a 0.1 mW He-Ne laser (λ = 632.8 nm) light source, focused through a 100x objective lens, and irradiated a SERS substrate to measure the Raman spectrum from 400-2000 cm-1 within 5 seconds 3 times.”

4, the function of EF is not correct. the function in the manuscript is only suit for the Raman test in solution.

Response:

We did perform SERS on solution samples. From the analytical chemistry point of view, the concentration of the solution instead of the number of probe molecules is known. Also, this concentration is directly proportional to the number of molecules. This equation was used in many SERS papers dealing with the enhancement factor. Please see the following references:

  1. ACS Omega 2021, 6, 41, 27271–27278.
  2. Journal of Food Measurement and Characterization volume 16, pages 2890–2898 (2022).
  3. J. Phys. Chem. C 2007, 111, 37, 13794–13803.
  4. Journal of Colloid and Interface Science, Volume 326, Issue 1, 1 October 2008, Pages 103-109.
  5. Talanta, 210, 2020, 120631.

  1. All pictures are low quality.

Response:

We have provided figures with higher quality in the revision.

Round 2

Reviewer 2 Report

I recommend this manuscript be published after minor modifications this time. 

1. For the title, the “High sensitive” should be “Highly sensitive”. 

2. For my last comment 14, about the enhancement factor (EF), the SERS and normal Raman of probe molecule (R6G) need to be measured. However, the selected R6G band at 612 cm-1 has been strongly interfered with by the TiO2 peak. I think no evidence shows the normal Raman of R6G has an intensity of about 108.6 a.u. If the authors cannot measure the normal Raman of R6G, I suggest a conservative estimate of the EF by considering the normal Raman of R6G is no higher than the signal noise level.

Author Response

Response to Comments and Suggestions

Reviewer 2

  1. For the title, the “High sensitive” should be “Highly sensitive”.

Response:

Thanks for reviewer’s suggestion. The title is revised.

  1. For my last comment 14, about the enhancement factor (EF), the SERS and normal Raman of probe molecule (R6G) need to be measured. However, the selected R6G band at 612 cm-1 has been strongly interfered with by the TiO2 peak. I think no evidence shows the normal Raman of R6G has an intensity of about 108.6 a.u. If the authors cannot measure the normal Raman of R6G, I suggest a conservative estimate of the EF by considering the normal Raman of R6G is no higher than the signal noise level.
  2. The caption of Figure 7 should be changed because the Raman spectra of the TiO2-deposited glass slide (A) cannot be called SERS.

(last comment 14. For the SERS enhancement factor (EF), the SERS and normal Raman of the same vibrational bands of R6G need to measure. However, in the data shown in Figure 8b, the normal Raman pattern does not show any R6G vibration bands. The so-called IRS (108.6) seems greatly contributed by the TiO2 band. In addition, the authors also confuse Raman with SERS in the caption of Figure 8. To avoid the interference of the TiO2 band, I suggest using a glass slide for the normal Raman measurement.)

Response:

Thanks for the reviewer’s comment. The normal Raman pattern of the glass slide is added to Figure 8b. We also have modified this Figure 8b caption accordingly. The added text on Line 286 is revised as “To compare the interference of the TiO2 peak, which has the same vibrational bands of R6G at 612 cm-1, the Raman pattern of the glass slide in our previous study is shown in Figure 8b [34]. Its IRS and EF values are 93.4 a.u. and 1.17 x 109, respectively. The interference can be neglected in the measurement of R6G.”

Reviewer 3 Report

 accepted 

Author Response

Thank you for accepting